

# The influence of diet on gut microbiome and body mass dynamics in a capital-breeding migratory bird

Isabelle Jones[1,*], Kirsty Marsh[1,*], Tess M. Handby[1], Kevin Hopkins[2], Julia Slezacek[3], Stuart Bearhop[1] and Xavier A. Harrison[1]

[1] Centre for Ecology and Conservation, University of Exeter, Penryn, United Kingdom
[2] Institute of Zoology, Zoological Society of London, London, United Kingdom
[3] Konrad Lorenz Institute of Ethology, University of Veterinary Medicine, Vienna, Austria
[*] These authors contributed equally to this work.

## ABSTRACT

Gut-associated microbial communities are known to play a vital role in the health and fitness of their hosts. Though studies investigating the factors associated with among-individual variation in microbiome structure in wild animal species are increasing, knowledge of this variation at the individual level is scarce, despite the clear link between microbiome and nutritional status uncovered in humans and model organisms. Here, we combine detailed observational data on life history and foraging preference with 16S rRNA profiling of the faecal microbiome to investigate the relationship between diet, microbiome stability and rates of body mass gain in a migratory capital-breeding bird, the light-bellied Brent goose (*Branta bernicla hrota)*. Our findings suggest that generalist feeders have microbiomes that are intermediate in diversity and composition between two foraging specialisms, and also show higher within-individual plasticity. We also suggest a link between foraging phenotype and the rates of mass gain during the spring staging of a capital breeder. This study offers rare insight into individual-level temporal dynamics of the gut microbiome of a wild host. Further work is needed to uncover the functional link between individual dietary choices, gut microbiome structure and stability, and the implications this has for the reproductive success of this capital breeder.

## INTRODUCTION

Gut-associated microbial communities play a pivotal role in organismal health and behaviour. Recent research has highlighted the importance of the gut microbiome for structuring host social networks (*Liberti et al., 2022*) influencing diet choice (*Trevelline & Kohl, 2022*) and shaping immunity and resistance to pathogens (*Belkaid & Harrison, 2017*; *Ley et al., 2008*). However, knowledge of the forces shaping the structure and stability of the gut microbiome remains limited, especially for non-model systems (*Bodawatta et al., 2021*). An understanding of the processes linked to plasticity or flexibility in the

Corresponding author
Xavier A. Harrison,
x.harrison@exeter.ac.uk

gut microbiome is vital for deriving accurate measures of the strength of host-microbe interactions in the wild, and the mechanisms that drive them.

The majority of gut microbiome research to date has focused on laboratory model systems including *Drosophila* spp., lab mice and animals of high economic importance such as poultry (*Abd El-Hack et al., 2022*; *Bodawatta et al., 2021*), or critically endangered species such as kakapos (*Strigops habroptilus*; *Bjerrum et al., 2006*; *Waite & Taylor, 2015*) and hummingbirds (*Dutch et al., 2022*; *Herder et al., 2021*). Studies such as these highlight how critical the gut microbiome is for host development and health (*Maki et al., 2019*) and experimental systems are well-placed to infer casual relationships between host and microbiome. However, the gut microbiome of wild species, especially birds, remains understudied (*Grond et al., 2018*) but is of vital importance in understanding the influence of host-microbiome interactions on host fitness, especially for particular taxonomic groups displaying unique life history strategies (*Bodawatta et al., 2022*; *Gil & Hird, 2022*; *Sun et al., 2022*).

Capital-breeding migratory birds, which use stored energy reserves to finance reproductive efforts, are a group where the consequences of microbiome variation are expected to be particularly pronounced. Due to their complex annual cycles, often involving use of multiple wintering and staging sites along the flyway, migratory birds are subject to a range of ecological pressures such as food availability or pathogen exposure that vary spatially and temporally, all of which have potential to shape the gut microbiome (*Harrison et al., 2013*). Migration is known to affect microbiome stability and therefore has the potential to shape host nutrition, metabolism, and energy yield from food (*Risely et al., 2018*). Work has identified the physiological reorganisation of organs pre-migratory departure (*Handby et al., 2022*; *Landys-Ciannelli, Piersma & Jukema, 2003*; *Piersma, Gudmundsson & Lilliendahl, 1999*). Which could be a potential driver of gut microbiota shifts. Many studies have focussed on characterising broad-scale patterns of microbiome variation at the group, site, or population level (*Pekarsky et al., 2021*; *Risely et al., 2017*). Though such group-level studies have revealed the gut microbiome of migratory birds to be highly variable in response to their environment and diet, they provide limited information on the underlying ecological processes that potentially drive this variation. Longitudinal measures of microbiome structure from individuals remain exceedingly rare for avian hosts because of the difficulty of repeat sampling individuals in the wild (*Videvall et al., 2018*; *Waite & Taylor, 2015*), yet could deepen our knowledge considerably of the role of gut microbial communities in the evolution and life history of a host (*Obrochta et al., 2022*; *Skeen et al., 2020*).

Here we use a capital-breeding migrant, the light-bellied Brent goose, (*Branta bernicla hrota;* hereafter LBB) as a model system for understanding the processes shaping the dynamics of the gut microbiome. Spring staging on the Álftanes Peninsula in Iceland during May is a critical period in the annual cycle where individuals sequester fat stores to finance the large energetic cost of migratory flight and subsequent reproduction. The reproductive success of the LBB goose is impacted not only by processes occurring during the breeding season, but also by those affecting mass gain during Icelandic staging (*Harrison et al., 2013*). The potential of these carryover effects has been the focus of previous studies

investigating the reproductive benefits gained by maximising body stores on the staging grounds (*Harrison et al., 2013*; *Inger et al., 2008*; *Inger et al., 2010*). Females who have the greatest mass at the end of the spring staging period have higher reproductive success within years where breeding conditions are favourable (*Harrison et al., 2013*).

Previous work in the wild has focused only on dietary quality as drivers of individual condition and has largely ignored the potential of the gut microbiome to modulate energetic yield from the diet (*Turnbaugh et al., 2006*; *Knutie, 2020*). The gut microbiome is known to play an important role in energy homeostasis by producing short-chain fatty acids, which are involved in various complex pathways regulating insulin resistance, adiposity, gluconeogenesis and satiety (*Besten et al., 2013*). In humans, reduced diversity and a higher ratio of the dominant Phyla Firmicutes and Bacteroidetes have been associated with obesity, although these patterns are not always consistent across studies (*Magne et al., 2020*). This variation among studies may be partly explained by differences in environmental factors and initial microbiome structure, as mammals have been shown to host different 'enterotypes' of microbiome which respond differently to factors such as diet (*Couch et al., 2021*; *Wang et al., 2014*; *Wu et al., 2011*). Variation in metabolic phenotype of the host gut, driven by differences in microbiome structure, could therefore generate asymmetries between individuals in the rate of energy assimilation even if they were foraging the same resource (*Turnbaugh et al., 2006*). This poses the question of whether the optimal foraging choice for individuals may be modulated by the host gut microbiome by controlling the rate of energy and nutrient intake from the host's diet (*Bäckhed et al., 2004*; *Trevelline & Kohl, 2022*).

The spring staging site inhabited by the geese in Iceland comprises two feeding resources; intertidal marine resources (*Zostera* spp., *Enteromorpha* spp. And *Ulva lactuca*) and terrestrial resources (maintained and agricultural grasses; *Inger et al., 2008*). Prior research into the LBB goose has found that individuals have a dietary phenotype with individuals either specialising on one resource or generalising on both resources (*Inger et al., 2006*). Individuals that specialise on marine resources have a higher average body mass and condition (*Inger et al., 2008*). These two feeding specialisations are bridged by generalist individuals which are switching between the two resources. For capital breeding species with strict migratory schedules and narrow temporal windows in which to attain a threshold body condition, the gut microbiome structure and the matching of gut microbiome enterotype to diet may be especially crucial (*Wu et al., 2018*). During this short, month-long spring staging period, feeding generalism could be costly if the microbiome has a period of adjustment to the new nutritional niche. Within this transition, energetic uptake could be compromised leading to a lower-than-average condition and therefore a mismatch with the phenology on the breeding grounds (*Inger et al., 2008*). We hypothesise that body condition and mass gain as a function of diet and foraging phenotype will be associated with differences in gut microbiome structure and diversity. We predict that: (i) foraging phenotype (marine specialist, terrestrial specialist, or 'switcher') will be linked to differences in the diversity and the structure of the gut microbiome; and (ii) foraging specialists will show higher rates of mass gain, because they are not paying the physiological cost of incompatible microbiomes; and finally (iii) marine specialists should have on average

better body condition at the end of spring staging than terrestrial specialists because of the superior nutrient quality of the resource (*Inger, 2006*). This work will improve our understanding of the selective forces facilitating the emergence and persistence of foraging specialisms. Portions of this text were previously published as part of a preprint (*Jones et al., 2023*).

## METHODOLOGY

### Data collection

Foraging preference, body condition and microbiome data of LBB geese were sampled during the spring staging of May 2017 on the Álftanes Peninsula, Iceland (64°06′27.5″N, 22°00′11.9″W). Individual geese were marked using coloured leg rings containing unique alphanumeric codes (*Harrison et al., 2011*). Observations of the geese across Álftanes were made over an approx. 12-hour period on a daily basis. The data were collected by researchers working in a pair, with one individual observing the focal goose at a distance of between 20–50 m using a high-powered telescope until the focal goose produced a faecal sample (as a proxy for distal gut microbial content; (*Videvall et al., 2018*). The observing individual then directed a second researcher to the faecal sample using two-way radios, with the samples then being placed into sterile plastic bags and kept initially at −20 °C before long-term storage at −80 °C prior to metagenomic sequencing. The individual collecting the samples was only dispatched when the observer could be confident that faecal sample could be unambiguously identified through the scope. The goose ID, time of day, flock size, abdominal profile index (API), and feeding location (marine or terrestrial sites) were all recorded. API is a reliable proxy for body condition (*Inger et al., 2010*) and was measured on a scale of 1–7, 1 being smallest and 7 being the largest abdomen which was protruding closer to the ground. Calibration of the API scores among researchers was carried out over the initial few days to ensure repeatability. Individual birds were sampled opportunistically each week throughout May, yielding 123 faecal samples, 1–4 samples per individual for sequencing, with a mean of 2.05. The two types of foraging specialisations were categorised based on location and flora present. The "marine" locations were salt marsh areas and contained *Zostera* spp., *Enteromorpha* spp. and *Ulva lactuca*. The "terrestrial" locations were managed grassland areas or agricultural grasses as well as saltmarsh. An individual's foraging specialisation was allocated based on the location of the bird when the faecal sample was collected as well as the location of the bird when it was resighted (for individuals observed at least five times). In total, 92 birds were observed five or more times and were therefore assigned a foraging phenotype. Individuals which were seen on both marine and terrestrial sites at least once were categorised as "switchers", while individuals seen only on one type of habitat were categorised as specialists on either marine or terrestrial resources. This work was reviewed and approved by the ZSL Ethics Committee (Project ID Code BPE-0682).

### 16S rRNA gene sequencing and bioinformatics

DNA extraction and sequencing of the v4 region of the 16S rRNA gene was conducted at the Institute of Zoology, London Zoo on the Illumina Miseq. We used a modified

Qiagen DNEasy plate extraction protocol with an added digestion step using mutanolysin to increase recovery of gram-positive bacteria (*Yuan et al., 2012*). Positive and negative controls were used. Extensive details of the extraction and 16S rRNA gene amplification protocol are provided in supplementary material of *Harrison et al. (2019)*. The raw 16S rRNA gene reads were processed in the DADA2 pipeline using default parameters to determine amplicon sequence variants (*Callahan et al., 2016*). The *phyloseq* package (*McMurdie & Holmes, 2013*) was used for downstream sequence processing in R (*R Core Team, 2021*). We removed 2081 chloroplast, mitochondria and archaea ASVs using the prune_taxa function (*McMurdie & Holmes, 2013*). Within the package *decontam* (*Davis et al., 2017*) a prevalence threshold of 0.5 was used to identify seven negative contaminant ASVs and these were subsequently removed. The *iNEXT* package was used to generate rarefaction and sample coverage curves to determine at what point diversity estimates plateau, this threshold was used to remove samples with less than 5,000 reads ($n = 22$; *Hsieh, Ma & Chao, 2016*; Figs. S1, S2).

## Statistical analysis
### Alpha diversity
Alpha diversity was calculated using richness estimated using the *iNEXT* package. A Bayesian regression analysis with richness as the response variable, and foraging phenotype as the explanatory variable, was fitted using the Stan computational framework with the R package *brms* (*Bürkner, 2017*) to examine which factors best explain variation in alpha diversity. Bird ID was fitted as a random effect to account for repeat observations of the individual geese. Day of year was standardised to have a mean of zero and standard deviation of one to aid model convergence. To assess how differences in group sizes may influence the patterns in alpha diversity across foraging phenotypes, the rarefied data was subsampled to the minimum group size with one random sample per bird ($n = 6$, marine specialists), and the average richness value calculated over 1,000 iterations to produce a distribution of bootstrapped richness estimates per foraging phenotype.

### Beta diversity
A centred log ratio (CLR) transformation of ASV abundances was used for beta diversity analyses, which does not require data to be lost to rarefying, accounts for the compositional nature of the microbiome data and normalises the read depth of the dataset (*Gloor et al., 2017*). A PERMANOVA was performed using Euclidean distances with the adonis2 function in the R package *vegan* (*Oksanen et al., 2020*). Pairwise comparisons were used to assess which foraging phenotypes differed in microbiome structure with the *pairwiseAdonis* package (*Martinez Arbizu, 2020*). We visualized variation in microbiome structure within and among foraging phenotypes using a Principle Component analysis performed on the CLR-transformed data. A test for dispersion was performed using betadisper in the *vegan* package (*Oksanen et al., 2020*) followed by a Tukey's *post-hoc* test of pairwise differences with a Bonferroni correction for multiple testing.

### Shared and unique taxa

To account for the uneven sampling effort across foraging phenotypes when calculating the proportions of shared and unique taxa at the group level, we performed bootstrap subsampling with replacement on the unrarefied data using the minimum sample size using one random sample per individual ($n = 6$, marine specialists) over 1,000 iterations. We randomly selected one iteration of the subsampling for visualisation. A bipartite network diagram was constructed using the R package *ggraph* (*Pedersen, 2021*), with foraging phenotype as one type of node, and ASVs as the second type of node. Edges were defined as an ASV being present in any sample from a given foraging phenotype. To test for differences in the proportion of shared taxa among different foraging phenotypes, we performed permutational *t*-tests of the pairwise Jaccard Index with 1,000 permutations between the following groups of sample pairs (removing within-individual comparisons); same phenotype (marine-marine/terrestrial-terrestrial), switcher-specialist (switcher-marine/switcher-terrestrial) and different specialists (marine-terrestrial). An indicator analysis was performed using the package *labdsv* (*Dufrene & Legendre, 1997*; *Roberts, 2016*). to determine the specificity and fidelity of ASVs to each foraging phenotypes. Set seed "17072020" was used to ensure reproducibility.

### Investigating drivers of body condition

To investigate the influence of foraging phenotype on body condition, we fitted a linear mixed model with API as the response variable, and foraging phenotype, day of year (and its quadratic effect), and sex as explanatory variables in the R package *lme4* (*Bates et al., 2015*). Bird ID was fitted as a random effect to account for pseudoreplication within the data. Day of year was standardised to have a mean of zero and standard deviation of one to aid model convergence. Competing models were ranked using AIC selection *via* the dredge function in the *MuMIn* package (*Barton, 2020*). All models within six AICc units of the best supported AICc model were considered to be relatively equally supported in the data. To remove overly complex models from consideration the nesting rule was applied (*Harrison et al., 2018*; *Richards, 2008*).

## RESULTS

### Alpha diversity

Richness differed among the three foraging phenotypes (Fig. 1A). Terrestrial specialists had the lowest microbiome alpha diversity compared to the other foraging phenotypes (marine *vs* terrestrial difference (95% Confidence Interval) = −0.71 [−1.05, −0.39]; switcher *vs* terrestrial difference (95% Confidence Interval) = −0.50 [−0.75, −0.25]). Marine specialists had the highest alpha diversity but was not significantly higher than that of switchers (switcher *vs* marine difference (95% Confidence Interval) = −0.21 [−0.55, 0.12]). Subsampling the foraging phenotypes to the minimum group size produced mean richness estimates that were similar to those predicted from the above model, with marine specialists showing the highest richness, switchers an intermediate value and terrestrial specialists showing the lowest richness (Fig. S3).

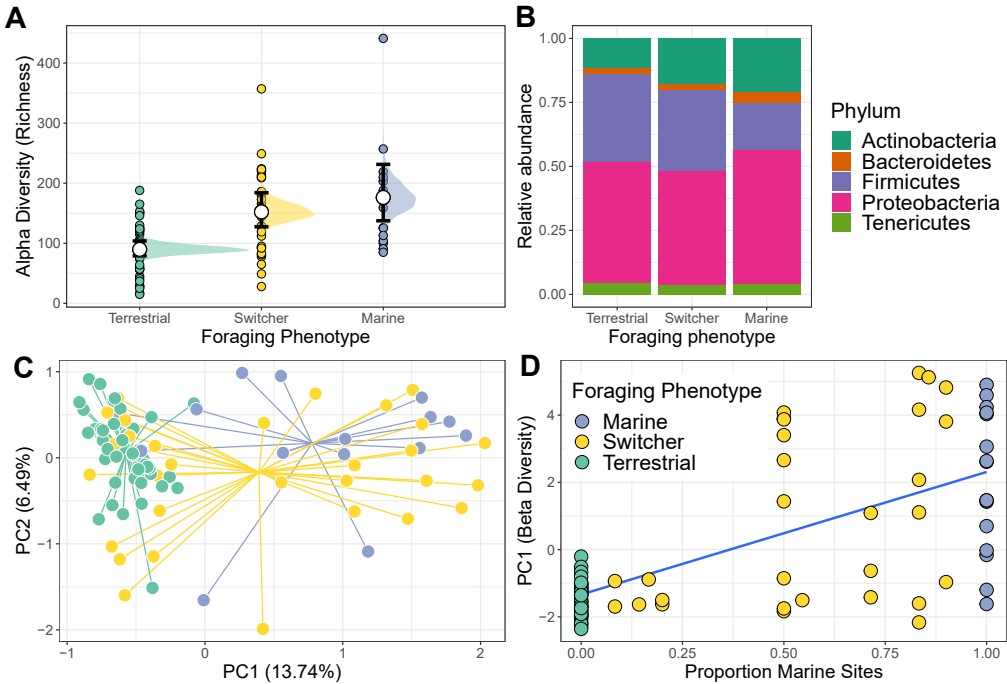

**Figure 1  Alpha and beta diversity of phoraging phenotypes.** (A) Variation in alpha (richness) diversity across different foraging phenotypes (Marine $n = 17$, Switcher $n = 34$, Terrestrial $n = 57$). Filled circles are raw data points. White circles represent posterior means from a Bayesian GLM. Black bars show 66% and 95% credible intervals of the means. (B) The relative abundance of the top five most abundant bacterial Phyla are shown for each foraging phenotype. (C) A principle component analysis on centred log-ratio transformed microbiome community data reveals Switchers to have a mean community composition intermediate to that of the two specialisms. Each point represents a sampled gut community and are coloured by individual foraging specialisation. Points are connected to group centroids by lines. (D) Values from the primary axis of the ordination in C plotted as a function of the proportion of marine sites used by individual geese ($n = 108$ geese). Line and shaded area are mean and 95% confidence intervals from a linear model.

## Beta diversity

The gut microbiome of individuals was primarily composed of the Phlya Actinobacteria, Bacteroidetes, Firmicutes, Proteobacteria and Tenericutes (Fig. S4). Gut community composition varied among the three foraging phenotypes (PERMANOVA; $F_{2,104} = 3.887$, $R^2 = 0.070$, $p = 0.001$) with marine specialists showing a lower proportion of Firmicutes but proportionally higher Proteobacteria compared to switchers and terrestrial specialists (Fig. 1B). The microbiome structure of terrestrial specialists clustered together along the first Principle Component axis (PC1; Fig. 1C) and were significantly different from that of other foraging specialists (PERMANOVA; marine *vs* terrestrial; $F_{1,67} = 6.210$, $R^2 = 0.085$, $p = 0.001$ and switcher *vs* terrestrial; $F_{1,88} = 4.092$, $R^2 = 0.044$, $p = 0.001$). Marine specialists and switchers showed more overlap in their composition, but a weak difference was still detectable between these foraging phenotypes (Fig. 1C; PERMANOVA $F_{1,53} = 1.528$, $R^2 = 0.028$, $p = 0.044$). The average position of individuals along PC1 was linearly correlated to the proportion of time spent on marine or terrestrial sites (Fig. 1D;

slope of proportion marine in diet 3.5 (95% CI [2.63–4.36])). In addition, the foraging phenotypes differed significantly in how variable the composition of their microbiomes were ($F_{2,104} = 17.541$, $p = 0.001$), with terrestrial specialists showing the least variability (difference in mean distance to centroid [95% Confidence Intervals]; Terrestrial-Marine; $-13.555$ [$-20.402, -6.707$], $p < 0.0001$, Terrestrial-Switcher; $-10.826$ [$-16.057, -5.596$], $p < 0.0001$) and marine and switcher phenotypes being equally variable (Switcher-Marine; $-2.728$ [$-9.880, 4.423$], $p = 0.637$).

## Shared and unique taxa among foraging phenotypes

Only four ASVs were consistently present across the majority of the gut communities sampled irrespective of foraging phenotype (using thresholds of 70% prevalence and 0.001% relative abundance); these belonged to Lactobacillaceae (*Lactobacillus*), Micrococcaceae, Aurantimonadaceae (*Aureimonas*) and Microbacteriaceae. The same was true when looking within each foraging phenotype, with Micrococcaceae only common among terrestrial samples, and the other three common within all foraging phenotypes.

We next examined the extent of shared and unique microbial taxa among foraging phenotypes. Strikingly, the marine foraging group had the highest proportion of unique ASVs (34.95% of total), followed by switchers (22.83%) and then terrestrial (14.86%) foraging groups as estimated through bootstrapping (Fig. 2A; Fig. S5). Marine and switcher foraging groups shared the most ASVs in common, with the smallest proportion of shared ASVs between marine and terrestrial groups (2.76%), and approximately 9.31% of ASVs common among all foraging groups (Fig. 2A). There were slight taxonomic biases across these subsets of the community, which broadly mirror the overall composition of each group. For instance, the ASVs unique or shared with terrestrial communities were enriched for Firmicutes while ASVs unique or shared with marine samples were enriched for Proteobacteria (Fig. 2B).

At the sample level, gut communities of individuals using the same habitat (marine-marine and terrestrial-terrestrial) shared significantly more ASVs in common compared to the proportion of ASVs shared between switchers and either marine or terrestrial specialists (observed $t = 29.93$, $p < 0.001$), or across the two different specialisms (marine-terrestrial; observed $t = 38.22$, $p < 0.001$). The switchers also shared significantly more ASVs in common with the two foraging specialisms compared to those shared between the different specialisms (observed $t = 15.87$, $p < 0.001$). In addition, terrestrial specialists shared more ASVs in common with each other than did marine specialists, while switchers shared the same number of ASVs in common with each other as they did with either foraging specialism (Fig. S6). Through an indicator analysis we identified four indicator ASVs for terrestrial specialists and two for marine specialists (see Table S1).

## Within-individual microbiome trajectory over time

Individual stability in microbiome structure differed among foraging phenotypes (GLMM of distances to individual centroids: $F_{2,38} = 7.801$, $p = 0.001$). Interestingly, we found that terrestrial specialist individuals had stable microbiomes through time, exhibiting relatively uniform values of both richness (Fig. 3A) and composition (PC1; Fig. 3B) throughout

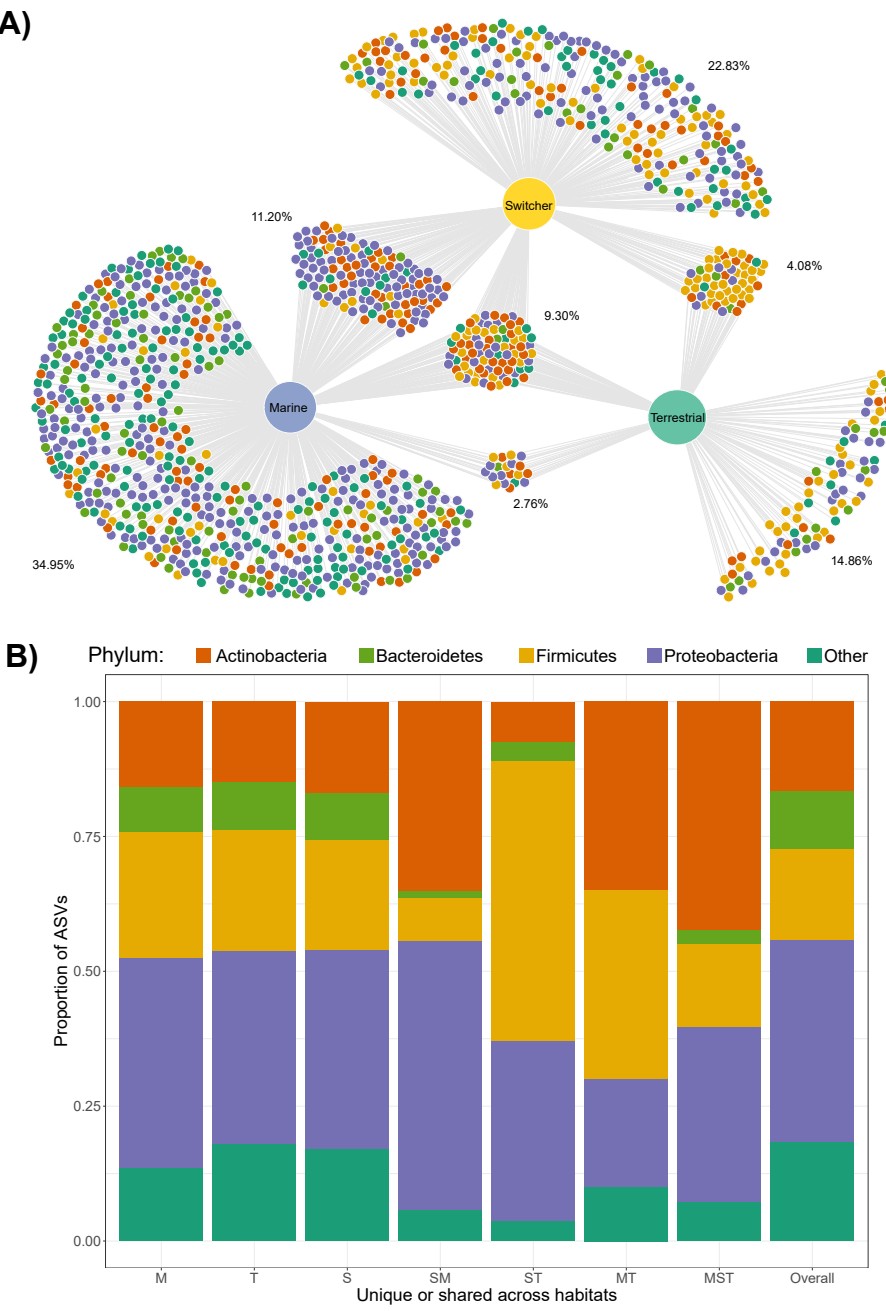

**Figure 2 Shared and unique microbial taxa among birds with different foraging phenotypes.** Taxonomy of microbial amplicon sequence variants (ASVs) at the Phylum level is indicated by the colours shown in the legend. (A) The ASVs shared and unique to Brent geese with different foraging preferences (marine, terrestrial or individuals switching between the two) are visualised in a bipartite network. ASVs are represented as nodes and are linked to the foraging phenotype of the samples they were present in. The average percentage of the ASVs per cluster of unique/shared ASVs estimated by bootstrapping are shown. (B) The taxonomic distribution at the Phylum level of each cluster of unique/shared ASVs in (A). M = marine, T = terrestrial, S = switcher, and combinations of these letters indicate clusters of shared ASVs. Samples from one randomly chosen iteration of a bootstrap subsampling routine ($n = 1244$ ASVs from $n = 6$ faecal samples per foraging phenotype) were used to visualise the network and taxonomic distributions of shared/unique taxa.

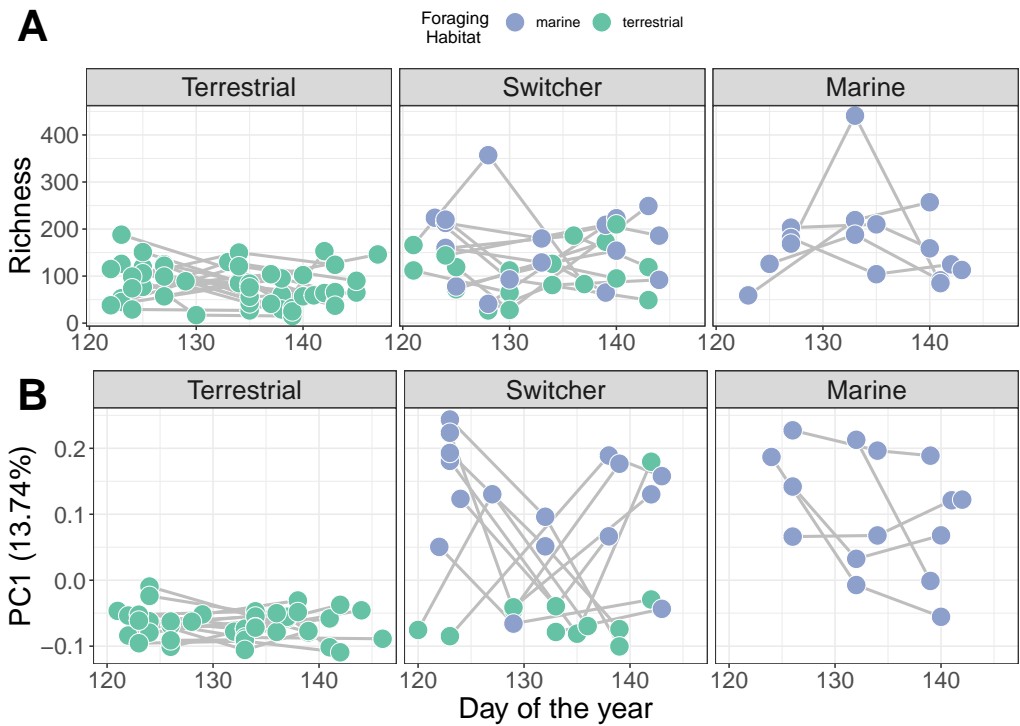

**Figure 3** **Microbiome trajectory by foraging phenotype over time.** Points represent faecal sample microbiomes with grey lines linking samples from the same individual host over time. Points are coloured by the classification of habitat the host was feeding on at the time of observation. (A) Alpha diversity (richness) over time (B) PC1 represents the first axis of a Principle Component Analysis performed on centred log-ratio transformed microbiome community data.

May. Consistent with group level dispersion patterns, switchers and marine individuals exhibited the most variation in microbiome structure over time (switcher *vs* marine: 3.805 [$\pm$3.700], $p = 0.310$; Fig. S7), while terrestrial specialists showed significantly lower variability compared to marine and switcher foraging phenotypes (terrestrial *vs* marine: $-11.973$ [$\pm$3.540], $p = 0.002$, terrestrial *vs* switcher: $-8.170$ [$\pm$2.704], $p = 0.005$; Fig. S6).

## Mass gain trajectories

The simplest model of variation in API retained foraging phenotype, sex, the interaction between standardised day and foraging phenotype, and its relevant quadratic term as predictors (Table S2). API increased significantly during spring staging as individuals gained mass prior to departure for breeding (z-standardised day; $p < 0.001$). In addition, there was a significant effect of sex on API over time, with females having a significantly higher API than males ($\beta \pm \text{SE} = 0.444 \pm 0.134$; $p = 0.002$). Consistent with our predictions, we also detected a significant interaction between z-standardised day and foraging phenotype type ($X^2 = 30.005$, $p < 0.001$). Switchers had the lowest rates of mass gain (Fig. 4), with terrestrial specialists having an API which increased significantly more over time than both marine specialists and switchers ($\beta \pm \text{SE} = 0.138 \pm 0.652$, $p = 0.034$). Interestingly, although these effects of foraging phenotype on mass gain trajectory are apparent in this

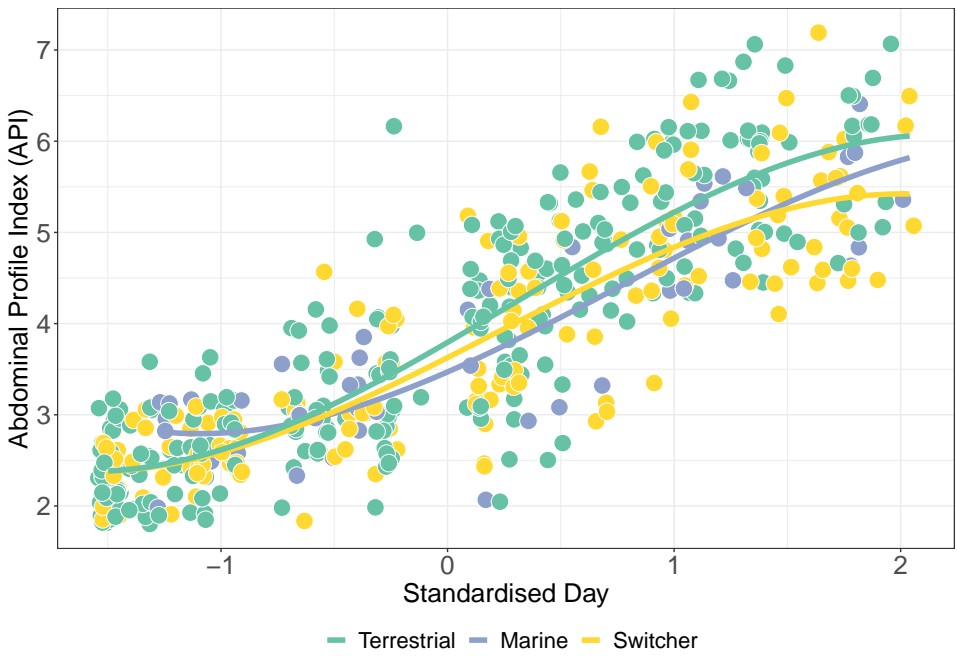

**Figure 4** **Mass gain trajectories over time.** Mass gain trajectories over time (measured as Abdominal Profile Index) for three foraging phenotypes, indicated by the colour of the points and trend line, with individual points jittered. The *X*-axis is standardised day of year (where 1st January = 1).

model, there remains a high degree of variation within these groups such that, for instance, switchers were amongst the individuals ending spring staging in both the best and worst body condition.

## DISCUSSION

Here we show how individual foraging behaviour is linked to the structure and stability of the faecal microbiota of LBB geese. Our study provides a rare example of within-individual longitudinal variation in the faecal microbiome in a wild migratory bird, where the microbiome shows foraging phenotype-associated temporal dynamics. These data suggest a possible link between microbiome, diet choice and body mass gain in a capital breeding bird, which has the potential to significantly influence downstream reproductive success (*Harrison et al., 2013*). These results have broad implications for our understanding of the interplay between gut microbiome, diet and behaviour in natural systems.

Our results suggest that individual foraging phenotype has a significant effect on the diversity, composition, and stability of the Brent goose faecal microbiome, and is linked to differences in body mass at the end of spring staging. Marine diets contain much less fibre than terrestrial grasses (*Inger, 2006*), making them more easily digestible and so marine specialists are perhaps less reliant on their gut microbes for efficient digestion of this resource. One might then expect to see reduced alpha diversity in hosts utilising marine resources due to a decrease in carbohydrate complexity and fibrolytic niche space,

coupled with an increase in variability among hosts due to weaker selective pressures and enhanced ecological drift (*Stothart et al., 2020*). While we did find higher dispersion among hosts with access to marine resources, the microbiomes of marine specialists and switchers were found to have higher alpha diversity than terrestrial specialists. This could be explained by higher diversity of (non-fibrous) dietary substrates, as marine diets comprise species such as eelgrass (*Zostera spp.*), and the algae *Enteromorpha spp.* and *Ulva lactuca.* However, the link between diet diversity and microbiome diversity in wild populations remains unresolved (*Bolnick et al., 2014*; *Kable et al., 2022*; *Kartzinel et al., 2019*; *Weinstein et al., 2021*). Alternatively, there could be a greater source of environmentally transmitted microbes on marine sites (*Grieneisen et al., 2019*; *Ottman et al., 2019*; *Zhou et al., 2018*). Less reliance on the microbiome for digestion may lead to increased uptake of transient environmental microbes, as predicted for avian hosts compared to mammalian counterparts (*Hammer, Sanders & Fierer, 2019*; *Risely et al., 2017*; *Song et al., 2020*).

The observed reduction in terrestrial specialist microbiome diversity may be due to exposure to chemicals on managed terrestrial grassland. Exposure to pesticides has been found to alter the honeybee (*Apis mellifera*) gut microbiome (*Kakumanu et al., 2016*). However, the impacts of habitat on the microbiome in wild birds have not been thoroughly studied (*Wu et al., 2018*). An experimental study found that urban diets reduced alpha diversity and altered taxonomic composition in house sparrows (*Passer domesticus*; *Teyssier et al., 2020*). Such environmental stressors could also explain the decrease in variability among terrestrial specialists, if they act as a filter for which microbes are able to tolerate these conditions (*Jani et al., 2021*; *Lavrinienko et al., 2020*). Therefore, it is possible that exposure to environmental stressors and pollutants associated with particular foraging strategies could be altering the gut microbiome of Brent geese. A study of migratory swan geese (*Anser cygnoides*) showed that gut microbial community structure and microbial interactions differed between breeding and wintering sites, which where differentially impacted by human activities (*Wu et al., 2018*). This could suggest that anthropogenic influences such as chemical exposure are playing a role in the differences in community structure observed in this study, as some terrestrial sites are managed golf courses maintained as monocultures with pesticides and fertilisers (*Obrochta et al., 2022*).

Switching individuals displayed signatures of faecal microbiome diversity and composition that were intermediate to both marine and terrestrial specialists, consistent with their gut microbiota existing in a transitionary state between the two diet-specific optima. Switchers showed the most dramatic variation in within-individual microbiome trajectory over time, reflecting a rapid microbial turnover in response to dietary shifts. A key hypothesis to be tested is that this constant microbial turnover incurs frequent mismatches between an individual's diet and the metabolic capacity of the microbiome to digest that diet. As a capital breeder with a narrow time window in which to reach peak mass, any changes in their mass gain capability could have major implications for their breeding success. We expected switcher faecal microbiomes to be an intermediate of the terrestrial and marine microbial communities, depending on proportional site use. Instead, switching individuals were found to have a high number of unique taxa (22.83% of the

total), which we would not expect to arise from sampling error alone. These microbes could represent certain taxonomic groups that allow switchers to digest food resources from both dietary types, or represents the fact that switchers are exposed to a greater diversity of environments, increasing the chance of environmental transmission of novel microbes. That marine and switchers shared the most ASV's in common could be due to the higher alpha diversity in marine individuals resulting in a larger probability of detecting an overlap in shared taxa.

All three foraging phenotypes showed large increases in API over spring staging, consistent with previous work on this system (*Inger et al., 2008*). Females also reached higher API values than males, as they require resources not only for the migratory flight but also investment into egg production (*Harrison et al., 2013*). Therefore, we expect diet—microbiome interactions and foraging behaviour to have sex-dependent consequences for the geese, with consequences of diet-microbiome mismatch felt more strongly in females. In contrast to previous work (*Inger et al., 2006*), we found that terrestrial specialists had significantly higher API gain over time compared to switchers and marine specialists. However, there was considerable variation in rate of API gain for all three foraging types. One hypothesis is that switchers could represent the socially inferior individuals oscillating between resource types because they are constantly being forced from high quality resources by more dominant individuals. However, this hypothesis alone cannot explain our data because some switchers end spring staging at some of the highest condition (API) scores. Instead, we think some switchers are better able to buffer the potential cost of switching through an ability of their microbiome to more rapidly shift metabolic capacity to match their diet, and/or may be moving among resources strategically to maximise foraging opportunities and energy intake. Similarly, switcher geese that change feeding phenotype at the start of the spring staging and then specialise on one diet for the remainder of the month could be finishing the month with higher API due to greatly reduced turnover of microbiome. Integrating detailed behavioural data alongside metrics of microbiome dynamics may be key to explaining the large degree of noise in API trajectory relationships. For example, geese from larger family groups have longer uninterrupted feeding because of reduced aggression from other geese and decreased vigilance behaviour (*Inger et al., 2010*). Such individual-level data are crucial for resolving why switchers show such variance in API at the end of staging and unravel whether 'decisions' to switch are driven by behavioural interactions such as competitive exclusion, or optimisation of resource intake and digestion.

## CONCLUSION

This study has highlighted the impact of feeding preference on body mass gain, potentially through shifts in gut microbiome structure, during a vital stage of a capital breeder's annual migration, with implications for among-individual variation in reproductive success in this wild host.

A major outstanding question is how shifts in the gut microbiota impact the ability of the host to uptake the maximum potential amount of nutrients from their food source. Our data reveal complex microbiome dynamics at the individual level over short timescales, but

also highlight that more intensive sampling is required to derive accurate metrics of rates of microbiome turnover and measure costs to the host. Though group level approaches can detect variation in *average* microbiome composition, they cannot estimate individual-level variation in community membership or rates of turnover that are so vital for being able to understand the *consequences* of microbiome variation for shaping animal life histories. Future research should also adopt a functional approach to the study of the microbiome (*i.e.,* using metagenomics, see *Gil & Hird, 2022*) to establish the link between diet-driven variation in gut community structure, metabolic function and body mass gained during the spring staging period. Understanding within-individual variance in the presence and function of gut microbial species over time will allow us to more clearly elucidate the role of the microbiome in driving variation in digestion, nutrition and condition in natural systems.

## ACKNOWLEDGEMENTS

The authors would like to thank Kerry Mackie, Kendrew Colhoun, Oli Torfasson, Sara Lupi and the wider International Brent Goose research group for support in the field.

### Funding

This work was supported by a Royal Society Research Grant (RG130550) and a Leverhulme Trust Research Grant (RPG-2020-320), both awarded to Xavier Harrison. The funders had no role in study design, data collection and analysis, decision to publish, or preparation of the manuscript.

### Grant Disclosures

The following grant information was disclosed by the authors:
Royal Society Research: RG130550.
Leverhulme Trust Research: RPG-2020-320.

### Competing Interests

Xavier Harrison is an Academic Editor for PeerJ.

### Author Contributions

- Isabelle Jones analyzed the data, prepared figures and/or tables, authored or reviewed drafts of the article, and approved the final draft.
- Kirsty Marsh analyzed the data, prepared figures and/or tables, authored or reviewed drafts of the article, and approved the final draft.
- Tess M. Handby performed the experiments, authored or reviewed drafts of the article, and approved the final draft.
- Kevin Hopkins performed the experiments, authored or reviewed drafts of the article, and approved the final draft.
- Julia Slezacek performed the experiments, authored or reviewed drafts of the article, and approved the final draft.

- Stuart Bearhop performed the experiments, authored or reviewed drafts of the article, and approved the final draft.
- Xavier A. Harrison conceived and designed the experiments, performed the experiments, analyzed the data, prepared figures and/or tables, authored or reviewed drafts of the article, and approved the final draft.

## Animal Ethics

The following information was supplied relating to ethical approvals (i.e., approving body and any reference numbers):

This work was reviewed and approved by the ZSL Ethics Committee (Project ID Code BPE-0682).

## Data Availability

The raw sequence reads are available at SRA: PRJNA936094.

The metadata and R markdown of all data processing and analysis is available at Github: https://github.com/xavharrison/GooseMicrobiomes2023

## Supplemental Information

Supplemental information for this article can be found online at http://dx.doi.org/10.7717/peerj.16682#supplemental-information.

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
