# Peer review of "The influence of diet on gut microbiome and body mass dynamics in a capital-breeding migratory bird"

_PeerJ, doi:10.7717/peerj.16682_

## Round 0.1 · original submission · Minor Revisions

As you will see both reviewers have suggested some details that need addressing in the manuscript with extra details required in the methods and also some thoughts about intra-individual variation.

Reviewer 1 ·

Basic reporting

This manuscript details a study on the relationship between diet, body mass and the microbiome of a species of migratory goose. It is well written and overall, a solid study. The field work needed to collect the samples in this was obviously a large amount of work and the analyses are thorough. The authors should be commended. My only major critique is that I would like to see more discussion on intra-individual variation. Also, there were several grammatical mistakes, most of which I noted in the comments below, but please double check manuscript for things like commas, periods, parentheses, etc.


Line 65 – Please explain what capital breeding means.

Line 84 – Recommend adding in (Branta bernicla hrota; *hereafter* LBB)

Line 91 – remove comma after effects

Line 125 – I’m not sure that incompatible is the correct word. Using terminology from line 116-118 perhaps transitioning microbiomes might be applicable?

Line 136 – The data *were* collected

Line 195 – remove comma after comparisons

Line 359-362 – Please explain what anthropogenic influences could be playing a role in the geese microbiota. Are the sample collection sites near a city?

Line 371 – add comma after mass

Line 395 – is ability referring to diet/foraging behavior? This is unclear

Line 400 – I would rephrase to say “may be” key to explaining…

Line 401-403 – this sentence seems incomplete. And is also missing a period at the end.

Figure 1A – the black bars showing Cis are not visible or at least not very clear in the figure

Figure 1B – I’d love to see a figure for each individual in a relative abundance plot, would be fine as a supplemental figure.

Figure 1D – in caption period needed after 108 geese).

Figure 2A – very clever visualization. What do the colors of the ASVs indicate? If it’s phylum that should be stated.

Figure 3 – It is unclear what the gray lines represent – Edit: I spent more time looking at this. At first it is not clear what is going on in this figure and therefore needs to be clarified. I think what exactly the gray lines are needs to be explicitely stated. Following that, the figure could be modified for clarity. The gray lines and dots could be small so it is more clear what the lines are connecting and which dots are from the same individuals.

Figure 4 – caption is missing a “)” somewhere, I think after Index

Experimental design

no comment

Validity of the findings

no comment

Additional comments

Relating to intra-individual variation:

One of the more interesting aspects of this study, in my opinion, is the repeated sampling of individuals. But I think more needs to be included about this in the text. In analyses, is an individual goose’s microbiome composition the average of each sample collected? Was each sample considered on it’s own, with bird as a random effect? When you subsampled foraging phenotypes were libraries subsampled regardless of host (so one host could potentially have up to four different samples in the subsampled group) or was it set to be only one sample per host? The introduction states “This study offers rare insight into individual level temporal dynamics” but this aspect overall seems to be lacking in the main text. I think additional information in the discussion on intra-individual microbial composition would significantly increase the value of the study

Reviewer 2 ·

Basic reporting

The work is professionally formatted, with clear sentence structure and well proof read work. There are only a few minor grammar errors present and the references are formatted to a consistent style. There are sufficient, relevant references and they are well cited in the text.
The manuscript is well structured, though the Introduction is currently referred to as a background. The raw data files have been shared as a supplementary file, and they are clearly organised. There is also a supplementary file containing extra tables and graphs that provide extra information for the manuscript.
The work has a clear, well developed rationale and the questions that are posed are answered within the text.

Experimental design

The work fits well within the wider scope of PeerJ and answers some very current questions with regards to bird microbiomes, and their potential impact on weight and survival relating to specialism. There is a clear gap in the literature in this area, and the potential value of the manuscript is well identified. One area that could be clearer, however, is the wider relevance of microbiome diversity and its functional relevance - particularly as this is central to the assessments of diversity. Build greater evaluation on the importance of diversity and specific microbial taxa to advance the relevance of the points being made.
Ethics are well considered and the ethical reference is provided. The work appears to have minimal ethics impact on the geese being studied.
Methods are generally well explained, but there remain queries with regards to parts of the observation. Please explain, for example, who engaged in the API (one author or many) and how reliability was assessed between authors. An example of the API scale would also improve repeatability. Equally importantly, please provide greater detail on when the geese observations of habitat took place (dates, times, etc). Please explain what was considered to be a specialist versus a switcher. If an animal was seen in a marine environment but was not feeding, was this a switcher? What were the requisites for categorisation for being a marine specialist, for example.

Validity of the findings

Overall, the study findings are meaningful and are well presented. There is some good depth of evaluation present in the discussion with regards to the implications of these results. However, as mentioned previously, the work would benefit from some greater depth of analysis with regards to the beneficial effects of microbiome diversity and functional relevance of specific taxonomic groups. Some greater evaluation of methodology is also necessary: how certain could you be that the animals truly were marine or terrestrial specialists, for example?
This said, the conclusion is clearly stated and provides a good summary of the work

Additional comments

Dear Authors,
Thank you for submitting this manuscript to PeerJ. This is an interesting and valuable work, and it fills in some gaps in the field of microbiome science. There remain however some gaps in the explanation of the methods, and these need to be filled to improve repeatability. Similarly, some greater analysis on the wider role of the microbiome in digestion and weight in geese would provide greater insight for readers. This said, this is a professional manuscript, and I look forward to seeing a revised version of the work.

Annotated reviews are not available for download in order to protect the identity of reviewers who chose to remain anonymous.

---

## Round 0.2 · accepted · Accept

I have assessed the revisions made by the authors and am happy that they have made all the changes requested by the reviewers, as such I believe the manuscript is ready for publication.